# Detecting Winter Wheat Irrigation Signals Using SMAP Gridded Soil Moisture Data

**Zhen Hao [1], Hongli Zhao [2,\*], Chi Zhang [1], Hao Wang [2] and Yunzhong Jiang [2]**

[1] Faculty of Infrastructure Engineering, Dalian University of Technology, Dalian 116034, China; haozhen@mail.dlut.edu.cn (Z.H.); czhang@dlut.edu.cn (C.Z.)

[2] Department of Water Resources Research, China Institute of Water Resources and Hydropower Research, Beijing 100038, China; Wanghao@iwhr.com (H.W.); lark@iwhr.com (Y.J.)

\* Correspondence: zhaohl@iwhr.com; Tel.: +86-136-8121-8530

**Abstract:** The southern part of the Hebei Province is one of China's major crop-producing regions. Due to the continuous decline in groundwater level, agricultural water use is facing significant challenges. Precision agricultural irrigation management is undoubtedly an effective way to solve this problem. Based on multisource data (time series soil moisture active passive (SMAP) data, Moderate Resolution Imaging Spectroradiometer (MODIS) normalized difference vegetation index (NDVI) and evapotranspiration (ET), and meteorological station precipitation), the irrigation signal (frequency, timing and area) is detected in the southern part of the Hebei Province. The SMAP data was processed by the 5-point moving average method to reduce the error caused by the uncertainty of the microwave data derived SM. Irrigation signals can be detected by removing the precipitation effect and setting the SM change threshold. Based on the validation results, the overall accuracy of the irrigation signal detection is 77.08%. Simultaneously, considering the spatial resolution limitation of SMAP pixels, the SMAP irrigation area was downscaled using the winter wheat area extracted from MODIS NDVI. The analytical results of 55 winter wheat samples (5 samples in a group) showed that winter wheat covered by one SMAP pixel had an 82.72% growth consistency in surface water irrigation period, which can indicate a downscaling effectiveness. According to the above statistical analysis, this paper considers that although the spatial resolution of SMAP data is insufficient, it can reflect the change of SM more sensitively. In areas where the crop pattern is relatively uniform, the introduction of high-resolution crop pattern distribution can be used not only to detect irrigation signals but also to validate the effectiveness of irrigation signal detection by analyzing crop growth consistency. Therefore, the downscaling results can indicate the true winter wheat irrigation timing, area and frequency in the study area.

**Keywords:** irrigation signal; SMAP; irrigation intensity; winter wheat

## 1. Introduction

Winter wheat is the main crop in the North China Plain (NCP). Due to the high irrigation demand of winter wheat, more than 70% of the irrigated water resources are used for winter wheat irrigation every year [1]. The increasing population has led to a corresponding increase in the demand for agricultural, industrial and domestic water in the NCP. The surface water resources are insufficient, and groundwater has become the main source of water for the NCP [2]. In recent decades, the overexploitation of groundwater has led to a significant decline in groundwater levels, which increases not only environmental problems but also the pressure on agricultural food production [3,4]. Groundwater is the main source of water for NCP agriculture irrigation. Long-term dependence on groundwater for agricultural irrigation has resulted in groundwater over-

exploitation, and agricultural water irrigation needs to be reduced; however, the sustainable of food crop production must also be ensured [5,6]. Timely and effective monitoring of irrigation water is of great significance for agricultural water management and water resources protection. The irrigation signal includes the time, frequency and area of irrigation. Irrigation time can be used to dynamically correct irrigation schedules, while irrigation frequency and area can be used for the estimation and dynamic monitoring of agricultural irrigation water use [7–10]. This study prepares to establish a model that can be used to detect irrigation signals and dynamically acquire irrigation information. The results of the irrigation signal will be used for the dynamic monitoring of agricultural irrigation water to achieve refined management of agricultural irrigation.

With the continuous development of remote sensing technology, more remote sensing data can be used for irrigation information detection [11–15]. Compared with traditional agricultural statistical methods, remote sensing has a wide range of multifrequency, high spatial and temporal resolution advantages and has been widely used in agricultural management [16–18]. Representative data sources include Moderate Resolution Imaging Spectroradiometer (MODIS), which provides 250 m, 500 m and 1 km resolution daily surface reflectance data. The richness of time series and improvement in remote sensing data spatial resolution has greatly improved the accuracy of irrigated area identification [19]. In recent research, the Normalized Difference Vegetation Index (NDVI) has been extensively used as an effective indicator for irrigated area recognition based on optical remote sensing data [19–21]. An analysis of the time-varying pattern of NDVI is the primary method for identifying irrigated and non-irrigated areas. In particular, wheat and maize are affected by irrigation, and their NDVIs will appear to be higher than other vegetation [20,22]. Although the identification method for irrigated areas has been comprehensive, this irrigated area extraction method based on optical remote sensing data is mostly used for long-term irrigated area monitoring to analyze trends in irrigated areas over multiple years. Chen et al. [23] proposed a method for detecting irrigation extent, timing and frequency based on MODIS and Landsat remote sensing data, which is an important irrigation property for understanding the sustainability of water resources in arid and semiarid regions. The irrigation signal detection method based on the visible vegetation index must model the daily scale data, and this method is more suitable for irrigation signal detection in regions with less cloud cover. Remote sensing images of areas with more clouds are likely to miss the critical period of irrigation signal detection due to cloud pollution. Moreover, in addition to the influence of image quality, the response of vegetation to irrigation is lagged, which increases the uncertainty of irrigation timing detection.

In addition to the method of identifying the irrigated area by using vegetation index information, the change in the wetness index can also be used to identify the irrigation signal [24]. Based on the SM being higher in the irrigated area than in the non-irrigated area, some researchers have identified irrigated areas based on different principles. Based on the MODIS enhanced vegetation index (EVI) and land surface water index (LWSI) ratio method, Peng et al. [25] introduced the variable EVI/LWSI threshold function to improve the detection ability of this method in different rice crops under mixed rice crop patterns (single-season rice, early-season rice, and late-season rice). Abuzar et al. [26] used vegetation and thermal thresholds derived from Landsat and Advanced Spaceborne Thermal Emission and Reflection Radiometer (ASTER) data to detect the irrigated area in an Australian irrigation district based on the soil temperature in the irrigated area being lower than that in the non-irrigated area. Although different researchers use SM information to detect irrigated area information from different aspects, they do not use SM indicators because optical and thermal infrared remote sensing data cannot directly obtain SM information.

Active and passive microwave satellites have proven to be effective tools for retrieving soil water variations at regional and global scales [27–29]. NASA's Soil Moisture Active Passive (SMAP) satellite, launched on 31 January 2015, provides a new source of data for near-surface (0–5 cm) soil water monitoring on a global scale. Colliander et al. [30] validated the SMAP surface SM product through the core validation site. The results indicate that the SMAP radiometer-based SM data product meets the expected performance of 0.04 m³/m³ volumetric SM (unbiased root mean square error) and that the combined radar-radiometer product is close to its expected performance of 0.04

$m^3/m^3$. Chan et al. and Zhang et al. [11,31] evaluated the results of different SMAP products in different regions and obtained similar conclusions to those of Colliander. SMAP has more information improvements than previous SM satellites, which has raised interest in whether SMAP can improve irrigation monitoring [32]. Subsequently, Lawston et al. [33] explored the use of SMAP data in identifying irrigation areas and timing in the Sacramento Valley, San Luis Valley and Columbia River Valley. However, the study did not identify the irrigation timing in the irrigated area. Since the detection of the irrigated area is a combination of changes in SM over a period of time, the time scale is the entire period of the crop. Compared with optical/thermal infrared methods, SMAP's method of detecting irrigated areas has unique advantages in terms of temporal resolution and ability to directly acquire SM [34]. The SMAP data spatial resolution is a major limiting factor that affects its use.

Obtaining irrigation time, area and frequency will help estimate irrigation water volume and provide data support for agricultural irrigation management. Despite having the low spatial resolution, SMAP provides high temporal resolution SM products. To address the spatial resolution issues, this paper will be studied in the following three aspects: 1) Based on SMAP and meteorological data, the irrigation signal in the study area was detected, which solved the problem of optical data not being applicable in cloudy regions; 2) MODIS remote sensing data were used to downscale the detection results to solve the low spatial resolution problem of SMAP data; and 3) through an analysis of the consistency of winter wheat growth covered by SMAP pixels, the SMAP data effectiveness in downscaling the winter wheat irrigation results in this study area was evaluated.

## 2. Study Area

The region of interest in this paper is located in the southern part of the Hebei Province and belongs to the NCP. The boundaries of the study area are city administrative boundaries, including Shi Jiazhuang, Baoding, Langfang, Hengshui, Cangzhou, Xingtai and Handan, with a total area of 8.9 $\times 10^4$ km² (as shown in Figure 1). Although precipitation in the study area is not scarce, the distribution of precipitation during the year is extremely uneven. The study area is dominated by a temperate monsoon climate with mean annual precipitation of 472.7–889.2 mm, and 70% of the annual precipitation occurs between June and September [35]. Under the irrigation conditions of the study area in recent years, the main crop pattern is the winter wheat-summer maize double crop rotation. Winter wheat and summer maize are also the main irrigated crops in this region [36]. The lower amount of precipitation in spring is not enough to provide sufficient water for winter wheat growth, and groundwater irrigation has been the main irrigation method for winter wheat and summer maize for a long time. Winter wheat is generally irrigated 4–5 times, and precipitation has little effect on the number of irrigations due to the severe shortage of precipitation during the winter wheat growing period. Summer maize is usually irrigated before planting, and if effective precipitation has occurred before planting and the soil moisture meets the sowing requirements, the crop will not be irrigated during the growing period. The Middle Route of the South-to-North Water Transfer Project (MSWTP) was launched at the end of 2014, and this project provided a new source of water for agricultural irrigation in the NCP [37].

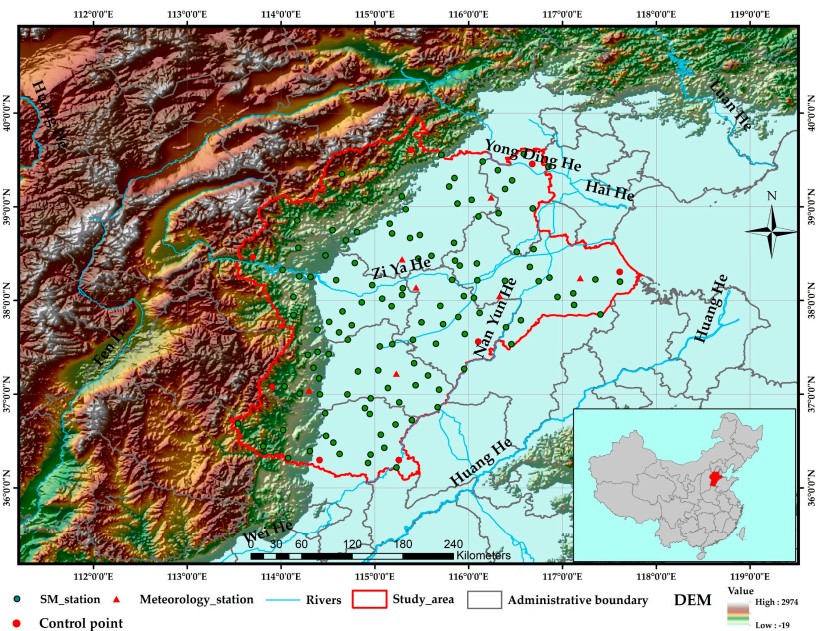

**Figure 1.** Study area and meteorological sites locations and the spatial distribution of SM stations.

## 3. Materials and Methods

The flow chart (shown in Figure 2) of this paper includes the processing of collected data (Section 3.1), selection of samples (Section 3.2.1), the application of algorithms (Section 3.2.2) and validation of accuracy (Section 3.2.3).

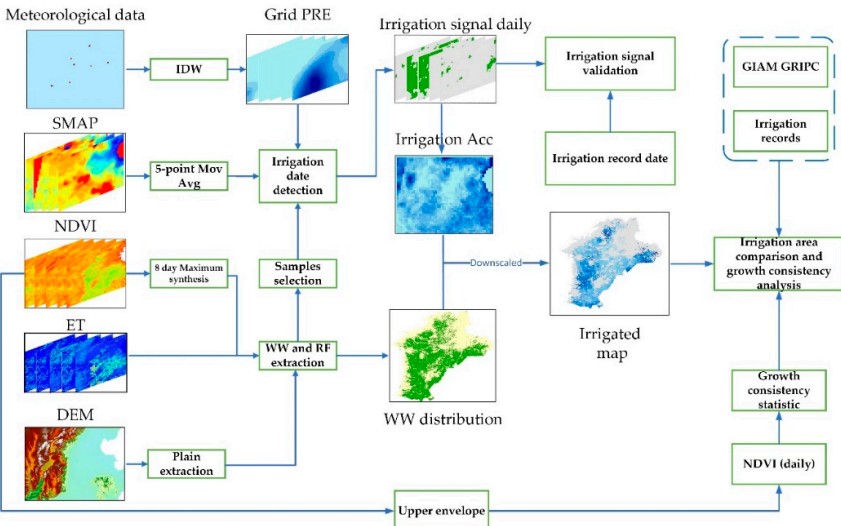

**Figure 2.** Flow chart for this study. Here, 5-point Mov Avg represents the 5-point moving average and Avg and Std represent the average and standard deviation, respectively. The irrigation Acc accumulates as a result of the irrigation signal.

### 3.1. Data Collection and Pre-Processing

#### 3.1.1. SMAP

SMAP is an orbiting observatory capable of measuring the amount of water in the top 5 cm of soil at global scales. To meet the various needs of soil moisture monitoring, the SMAP mission uses an L-band radar and an L-band radiometer for concurrent, coincident measurements integrated into

a single observational system and ultimately produce a variety of spatial and temporal resolution SM products [38]. Since the successful launch of SMAP in January 2015, it has provided SM products of many levels worldwide. After validating the accuracy of SMAP products, the SMAP products meet the mission requirements and can also be used to assess hydrologic processes [30,31,39]. In this study, considering the spatial and temporal resolution of the SMAP products required for the study area, SMAP enhanced L3 radiometer global daily 9 km EASE-grid soil moisture version 1 was used as the final data source [40]. The study used the SMAP SM datasets from the end of Feb to the end of May (2015–2018) corresponding to the winter wheat irrigation period in study area. Although the SMAP dataset can provide daily SM products, due to satellite orbits, SM products do not cover the spatial extent of the study area every day due to satellite orbits. To select the SMAP data that can completely cover the study area, 8 control points are used to filter the data that meet the requirements. The eight control points are evenly distributed at the vertices of the study area boundary, and the judgement equation is as follows:

$$X_i = \begin{cases} 1,\ 0 < V_i < 1 \\ 0,\ V_i = null \end{cases} \tag{1}$$

$$J = \begin{cases} R, \sum_i^8 X \geq 6 \\ D, \sum_i^8 X < 6 \end{cases}. \tag{2}$$

where X is the judgement result of the SMAP pixel value (*V*) and null is no-data in this pixel, i is the number of the control point. If the value of the SMAP pixel is between 0 and 1, X = 1; and if the SMAP pixel value is null, then X = 0. J is the judgement result of whether the SMAP data are retained, and R and D represent the retention and deletion of SMAP data, respectively. If the sum of the 8-control point X ≥ 6, it indicates that SMAP data can cover a large area (more than 75% of the study area is covered) of the study area and this SMAP data is retained; if it less than 6, the data are deleted. The programming language for batch filtering, processing and extracting of SMAP data is python 2.7, and the arcpy function provided by ArcGIS 10.4 (Environmental Systems Research Institute in California) is also used. Regarding the extraction of pixel values in this paper, the "ExtractValuesToPoints" function in arcpy is used.

### 3.1.2. MODIS

MODIS provides researchers with stable, long time series global remote sensing data. Some global land use/land cover (LULC) datasets based on MODIS data have been generated [41,42]. MOD09GA and MOD16A2 provide daily surface reflectance with a spatial resolution of 500 m and evapotranspiration of 500 m every 8 days [43,44]. MOD09GA and MOD16A2 were used in this study for irrigated area downscaling, while the latter is based on 8-day synthetic data and does not require further processing. MOD09GA is daily surface reflectance data, and cloud pollution has a large impact on the use of data. First, the NDVI is calculated based on the MOD09GA dataset.

$$NDVI = \frac{NIR - RED}{NIR + RED} \tag{3}$$

where *NIR* and *RED* are the surface reflectance factors for the presented wavelengths. Second, the 8-day maximum value composite (MVC) method was used for the daily NDVI time series dataset, and the method is to composite a new NDVI image by using the daily maximum value of NDVI within 8 days of each pixel in the image as a valid pixel value [45]. This processing method reduces the impact of clouds on the dataset and keeps the time resolution of the two MODIS products consistent. The batch redefinition projection and raster attribute conversion of MODIS data are based on MRT (MODIS Reprojection Tool supported by NASA, referenced by Dwyer et al. [46]), and the maximum synthesis of the NDVI is based on MATLAB 2018b.

### 3.1.3. Precipitation

The National Meteorological Information Centre of China provides daily precipitation data (meteorology station) from 1961 to present [47]. There are 2472 meteorology stations in China, and

there are 7 meteorological stations in this study area. Daily precipitation data were collected from March 2015 to December 2018. Since precipitation data must be coordinated with SMAP data for irrigation information monitoring, the spatiotemporal resolution of the precipitation data must be processed. The daily precipitation data include statistical results for two periods (20:00–8:00 and 8:00–20:00) in Beijing time. The current method for the L3_SM product is to use the nearest 6:00 AM local solar time criterion to perform Level 3 compositing [38]. The precipitation from 8:00–8:00 is summed as the daily precipitation, and the station data are interpolated into the grid data using the inverse distance weighting (IDW) method based on python 2.7.

### 3.1.4. Irrigated Map

A global irrigated area map (GIAM) and global rainfed, irrigated, and paddy cropland (GRIPC) were also collected in this study for irrigated area validation. Based on the unsupervised classification method, GIAM provides irrigated area recognition results with a spatial resolution of 1 km in the year 2000 [34]. GRIPC is the result of the decision tree method for the classification of MODIS data and the spatial resolution is 500 m in year 2005 [17].

### 3.1.5. In Situ SM Measurement Data and Irrigation Records

There are 135 SM stations in the study area, and SM data are provided every 10 days (1st, 11th and 21st). The SM data measurement (oven-drying method) depths include 10 cm, 20 cm and 40 cm, and the measurement time is concentrated at 8:00 AM Beijing time. These sites also provide information on precipitation and irrigation times between measurements. The recorded irrigation data include areas of agricultural irrigation, irrigation crops, timing and volume. Although the recorded irrigation information is relatively abundant, the spatial scale is the agricultural irrigation region. The data collected in this study are shown in Table 1. Since this study only collected information on irrigation records in 2018, only the SM changes in 2018 were plotted during sample training and validation.

**Table 1.** Datasets collected in this study.

| Data Source | Temporal Resolution | Spatial Resolution | Time Period | Data Access |
|---|---|---|---|---|
| SMAP | daily | 9 km | March 2015 to December 2018 | https://nsidc.org/data/SPL3SMP_E/versions/2 |
| PRE | daily | site | March 2015 to December 2018 | http://data.cma.cn/ |
| MOD09GA | daily | 500 m | March 2015 to December 2018 | https://ladsweb.modaps.eosdis.nasa.gov/ |
| MOD16A2 | 8-day | 500 m | March 2015 to December 2018 | https://ladsweb.modaps.eosdis.nasa.gov/ |
| Irrigated Map | year | 1 km and 500 m | | http://www.iwmi.cgiar.org/ https//dl.dropboxusercontent.com/u/12683052/GRIPCmap.zip |
| Irrigation Records | 10-day | site | January 2018 to December 2018 | |

PRE: precipitation.

### 3.2. Methods

#### 3.2.1. Established SMAP Training Samples for Winter Wheat and Rainfed Crops

The selection of training samples is important before establishing a model of irrigation signal detection. Since the SMAP data have a low spatial resolution, the training samples should be selected to ensure that the surrounding crops are consistent. In this paper, samples were selected using a

combination of MODIS NDVI and MODIS ET. Since the winter wheat (WW) NDVI in the NCP was significantly higher than other crop in March, the spatial distribution of WW can be extracted based on the March NDVI data. However, this spatial distribution may include other vegetation with a higher NDVI (such as landscape forest), and ET is needed to improve the extraction accuracy of WW. Since March to April is the main irrigation period for WW, the cumulative ET value of WW is significantly higher than that of other vegetation during this period [48]. This indicates that the extraction accuracy of WW can be improved by adding ET as a limiting condition. Using these two features (NDVI and ET), the WW pixels can be extracted more accurately. WW and rainfed crop pixels were extracted by the decision tree model in Figure 3.

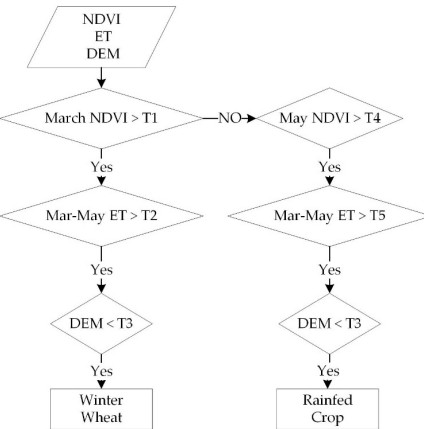

**Figure 3.** Winter wheat and rainfed crops planting area extraction model. Where March NDVI and Mar-May ET represent the NDVI in March (May NDVI is similar to March NDVI) and cumulative amount of ET from March to May, respectively; DEM is the elevation information; and T is the threshold in different conditions. If the pixel value (such as NDVI and ET) satisfies the threshold, the pixel value is 1, and if it is not satisfied, the pixel value is 0.

The selection of WW samples should be based on SM sites, and more irrigation information can be obtained. Rainfed crop samples should ensure that there are no irrigated crops nearby as much as possible, which can reduce the influence of surrounding crop irrigation on SM. Finally, 11 WW samples and 7 rainfed crop samples were established in the study area, 7 WW samples and 4 rainfed crop samples were used as training samples, and the remaining samples were used as validation samples. These samples are distributed from north to south and can reflect the difference in irrigation time of winter wheat under different latitude conditions.

### 3.2.2. Irrigation Information Detection and Irrigated Area Downscaling

Extracting the precipitation and SMAP time series data of the meteorological site spatial location can not only be used to evaluate the sensitivity of the SMAP data to the precipitation response but also to support the threshold setting of the irrigation signal detection. The irrigation signal detection is based on the SMAP SM variation. It can be assumed that if the SM of SMAP is increased and the grid has no significant precipitation, the increase in SMAP SM is caused by irrigation. Since the amplitude increase in the SMAP original SM signal is significant, it is difficult to detect irrigation by threshold segmentation and the original signal needs to be processed using the moving average method. In the original SMAP data, due to the existence of signal noise, the SM is may be suddenly reduced (previously without precipitation and irrigation), if this value is calculated with the SM at the latter time, the identified irrigation signal is invalid. SM Value in that time need to be corrected. To reduce the influence of SMAP SM data amplitude on the irrigation signal detection, a 5-point moving average method is used to process the SMAP SM original signal. The 5-point moving average not only ensures the amplitude of the original but also reduces the frequent fluctuations in the original signal. Sun et al. [49] compiled the water requirement for different growth stages of WW in

the NCP. In this paper, the daily precipitation of >4 mm was used as the threshold for effective precipitation (referenced by Sun et al.). In this study, the irrigation identification results of the grid were binarized (irrigation is 1, no irrigation is 0).

By accumulating the binarized daily irrigation identification results, the frequency of irrigation in the WW planting region can be obtained. Notably, the irrigation frequency of a grid may be higher than 6 times because the grid (9 km×9 km) cannot be completed irrigated in one day. After integrating the spatial distribution of the irrigation intensity and the WW planting area, the irrigated area with the irrigation intensity identification was finally obtained. However, the accuracy of the irrigated area recognition results based on a single SMAP data source does not meet the general application requirements. By introducing the previously extracted WW spatial distribution, the downscaled results of irrigation intensity were obtained from the SMAP irrigation intensity results without the influence of non-irrigation pixels (such as rainfed crops and city). The mathematical expression of the method in this section is as follows:

$$IS_{i,j} = SM_{i,j} > T_6 \text{ and } Pre_{i,j} < EPre \tag{4}$$

$$II_{i,j} = \frac{\sum_1^t IS_{i,j}}{\max(\sum_1^t IS_{m,n})} \tag{5}$$

$$II_{downscale} = \begin{cases} II_{i,j}, WW = 1 \\ 0, WW = 0 \end{cases} \tag{6}$$

where *i* and *j* represent the pixels of the ith row and jth column, respectively; *IS* is the irrigation signal; *SM* is the soil moisture derived from SMAP; $T_6$ is the threshold for soil moisture increase; *Pre* and *EPre* represent precipitation and effective precipitation, respectively; *II* is the irrigation intensity; *t* is the total number of days in the study period; $\max(\sum_1^t IS_{m,n})$ represents the maximum value of the accumulated value of the irrigation signal over the entire event range; and *II_{downscale}* is the downscaled irrigation intensity. In equation 6, the WW spatial distribution and the irrigation intensity image need to be calculated. If the WW spatial distribution image pixel value is 1, the $II_{downscale}$ pixel value is assigned as the irrigation intensity value. The irrigated area is calculated as the area of the pixel where the irrigation intensity is greater than zero. The algorithm implementation in this section still needs to use the arcpy function based on python 2.7.

### 3.2.3. Validation and Consistency Analysis

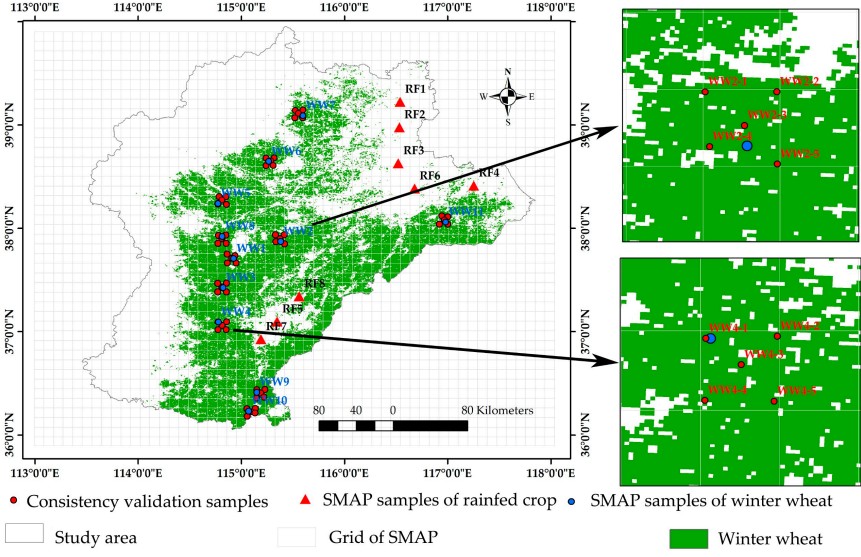

**Figure 4.** Sample maps. Red triangles and blue points are used to extract the SMAP SM time series signals from different crops; red points are used to extract the winter wheat NDVI time series signal and then compare the consistency of winter wheat growth covered by one SMAP pixel.

The results of the irrigation signal detection have been validated, and the uncertainty in the irrigated area downscaling has also been analyzed. First, the detection results of the irrigation signal are based on the irrigation record. Since the SMAP SM time series data in this paper used the 5-point moving average method, if the detected WW irrigation signal is different from the irrigation record in three days, the result is correct. Simultaneously, if the non-WW planting area also detects the irrigation signal, it is necessary to reset the irrigation signal detection threshold according to the irrigation signal frequency. The equation for the validation of irrigation timing is as follows:

$$Accuracy = \left(\frac{CDet}{ADet + WRec}\right) * 100\% \tag{7}$$

$$OA = \text{avg}(Accuracy_1 + Accuracy_2 + \cdots + Accuracy_l) \tag{8}$$

where *Accuracy* is the sample validation accuracy; *OA* is the overall accuracy and *l* indicates the total number of validation samples; *CDet* indicates the number of days that were correctly detected in the irrigation record; *ADet* represents the number of days for all irrigation detected results; and *WRec* is the number of days that have not been detected in the irrigation record.

Second, when using the WW data extracted by MODIS to downscale the SMAP irrigation signal detection results, whether the growth of WW covered by one SMAP pixel is consistent must be considered. The selection strategy for the consistent analysis of WW growth is to establish samples in four corners and center points covered by one SMAP pixel as shown in Figure 4. The NDVI daily signal extracted from the samples was subjected to upper envelope processing [23], and the signal was divided according to the growth stage of WW and the change in SM. By counting the number of samples from the consistent growth of WW, the consistency analysis results of WW growth covered by one SMAP pixel were obtained. The consistency analysis results are calculated as follows:

$$P = \left(\frac{RG + J}{10}\right) * 100\% \tag{9}$$

where *P* is the percentage of growth consistency of WW; *RG* and *J* are the number of consistent samples of WW growth in the returning green and jointing stages, respectively; and 10 is the number of samples for all these two stages. Five growth consistency samples can be obtained for each growth stage (corresponding to the red sample point), and 10 consistency analysis samples can be obtained for the two stages of the returning green and jointing stages.

## 4. Results and Validation

### 4.1. Irrigation Signal Detection

Taking four meteorological stations as examples, the time series of NDVI (8-day maximum synthesis), ET (8-day), precipitation and SM from 2015 to 2017 were plotted in Figure 5. Comparing the time series data of the four meteorological stations, it was found that the vegetation coverage of the Baoding and Nangong stations were rainfed crops and those of the Botou and Raoyang stations were WW. An analysis of the time series changes of NDVI and ET showed that the meteorological stations with WW vegetation cover (Botou and Raoyang) not only had more NDVI peaks than rainfed crop stations (Baoding and Nangong) but also significantly higher ET from March to May. Time series changes of precipitation and SM provide an important basis for irrigation signal detection. During the main growth period of WW (March to May), Botou and Raoyang stations were affected by irrigation and still maintained high SM without precipitation. Simultaneously, the SM observed in the WW growing season was more stable and higher than that of the non-irrigated crops.

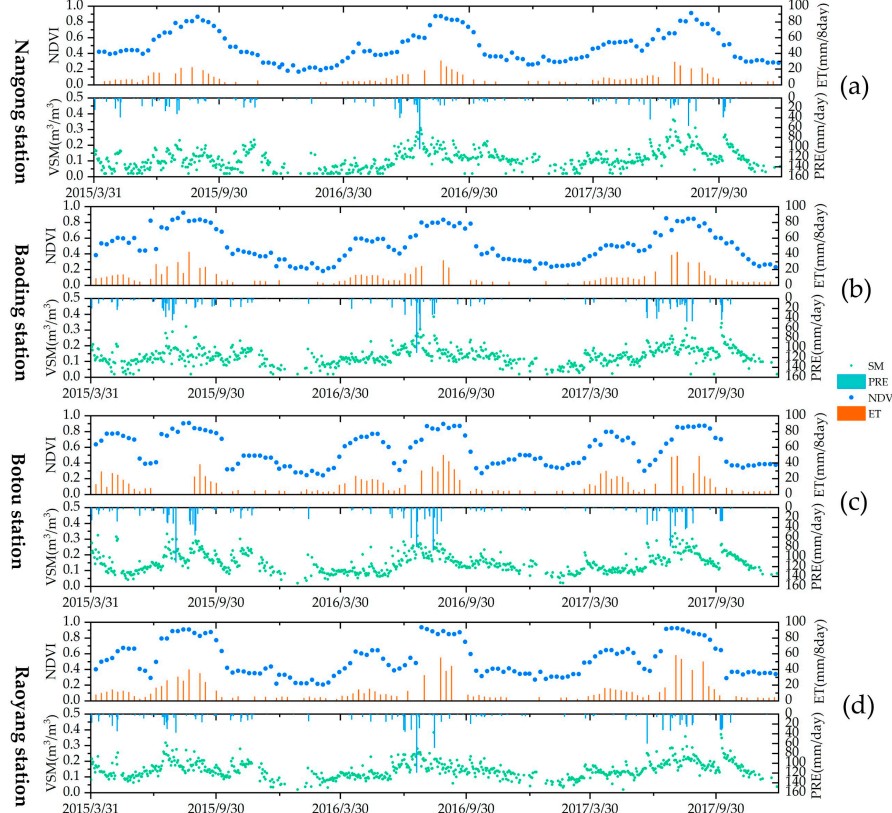

**Figure 5.** NDVI (8-day maximum synthesis), ET (8-day), precipitation (daily) and SM (daily) time series variations. (**a**) Nangong, (**b**) Baoding, (**c**) Botou and (**d**) Raoyang meteorological stations; and VSM means volume of soil moisture. The land cover at Nangong and Baoding stations was rainfed crops, and the land cover at Botou and Raoyang was winter wheat.

Using the 5-point moving average method for statistical time series SM results, which can reduce the influence of abnormal points on the irrigation signal detection. The smoothed SM results are shown in Figure 6. Figure 6a Changes in SM (blue lines) and effective precipitation events (green lines) in WW samples, and the statistical WW irrigation time is also plotted (Triangle point). Figure 6b Changes in SM and effective precipitation events for rainfed crops. The figure can reflect the response relationship between SM and precipitation, at the same time, by comparing the SM curves of different crops, it can be found that show the WW pixels have a more obvious SM increase than rainfed crop pixels. Comparing WW samples with rainfed crop samples, it was found that both had an increasing trend in SM before the first recorded irrigation. The slowly increasing trend in SM under no precipitation conditions may be caused by seasonal and vegetation water content changes [31]. However, the increasing trend in WW samples with different spatial locations was different before the first irrigation stage. Due to the difference in temperature, the irrigation time was different. The SM of the WW sample in the southern region increased significantly compared to the WW samples in the northern region (top line in Figure 6a is the southern region WW sample, and the bottom is the northern region). Both WW samples and rainfed crop samples have significant SM increase feedbacks under effective rainfall events. The difference is that irrigation events will also significantly increase SM without effective rainfall, which is shown in Figure 6. Setting the threshold for SM increase without an effective rainfall event can be used to detect irrigation signals in the WW region.

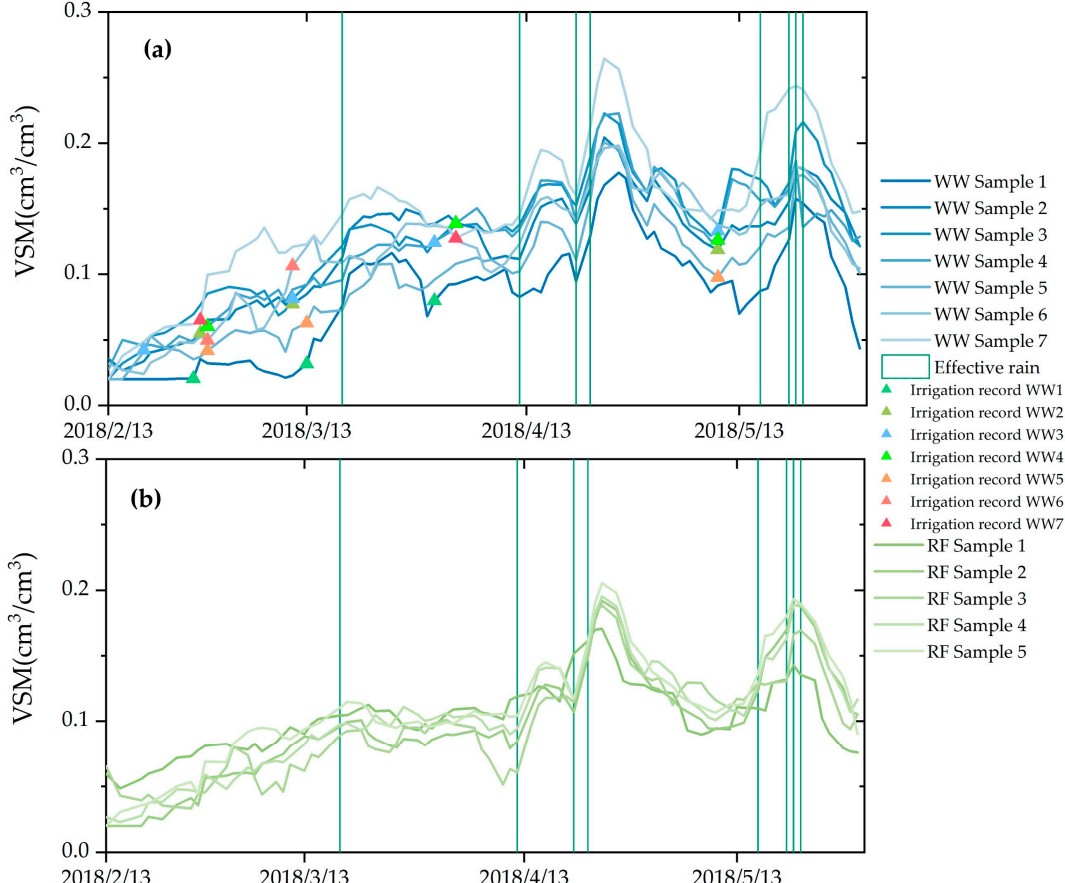

**Figure 6.** Training samples of irrigation signal detection. (**a**) Winter wheat training samples, and (**b**) rainfed crop training samples. The irrigation record is a summary of the irrigation records of the main irrigation region in the study area and used as a reference for the water supply time for winter wheat.

The irrigation signal detection results of WW and rainfed crops are shown in Figure 7a,b, respectively. By setting the SM change threshold, the time when the SM was significantly increased without effective precipitation is detected as the irrigation time (square point in Figure 7). In the rainfed crop region, only one irrigation signal was detected in this region due to the setting of the SM increase threshold. By comparing the SM trend of WW and rainfed crops, the SM trend in the WW region was more obvious, and there was also a significant increase (it is affected by irrigation) in SM when there was no precipitation. The SM trend in the rainfed crop region is more stable. Under the same precipitation conditions, the SM increase in the rainfed crop region is lower than that in the WW region. According to the results of WW irrigation signal detection, the irrigation frequency was higher from mid-February to mid-March. Due to the high frequency of precipitation in April and May, the irrigation frequency is lower than in February and March. Additionally, in the early WW growth stage (turning green and jointing), the main irrigation water source in the study area is surface water, and the amount of irrigation water will be more than that in the middle and late growth stages of WW. For different study areas, the setting of effective precipitation can be stricter, which may reduce the false detection of irrigation signals. Notably, the results of irrigation signal detection in this paper were large-scale surface water irrigation signals. Due to the small amount of irrigation water and the dispersion of irrigation areas, SMAP pixels do not easily reflect changes in SM amplitude caused by groundwater irrigation.

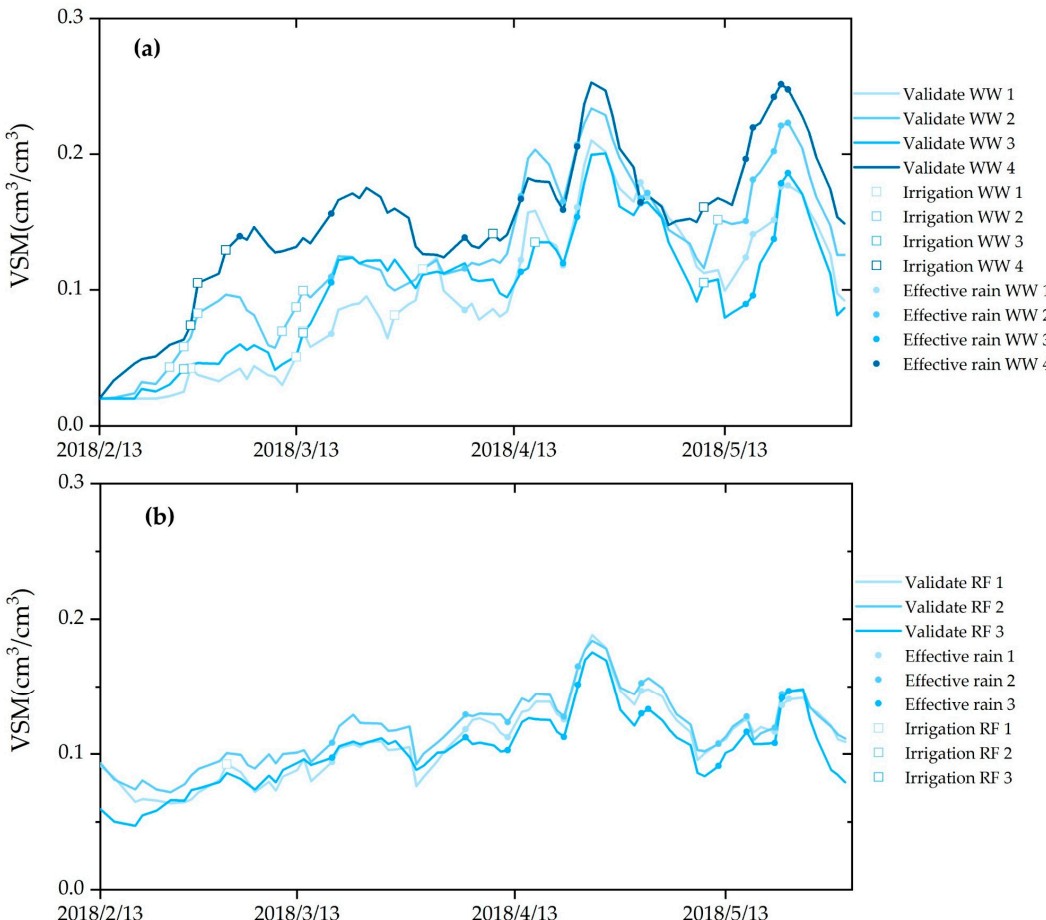

**Figure 7.** Irrigation signal detection results. (**a**) WW sample detection result and (**b**) RF samples detection result. The time corresponding to the square mark is the irrigation time, and the time corresponding to the circle mark is the effective rain time.

*4.2. WW Extraction Results and Irrigated Area*

In this paper, irrigation signal detection training samples must refer to both WW and rainfed crops. Figure 8a,b were obtained by daily NDVI using an 8-day maximum synthesis process, and Figure 8c was the cumulative ET from early March to early May. According to the crop growth phenology of the study area, only the WW crop in the study area showed obvious vegetation characteristics in March and early April. Therefore, most of the green areas in Figure 8a characterize the spatial distribution of WW. Since WW is already irrigated, the cumulative ET is significantly higher than that of other crops. Combined with the cumulative ET in Figure 8c, WW pixels with higher precision can be extracted. The vegetation characteristics of rainfed crop pixels appeared later than that of WW, and the cumulative ET was significantly lower than that of WW.

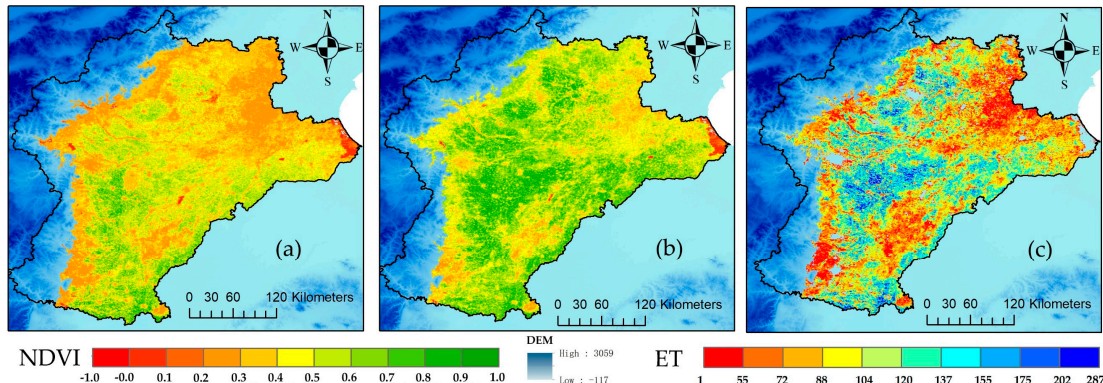

**Figure 8.** Sample selection based on MODIS NDVI and ET: (**a**) MODIS NDVI of DOY (day of year) 89-97, (**b**) MODIS NDVI of DOY 116-124, (**c**) MODIS ET accumulate from DOY 65-129.

The normalized results of the cumulative irrigation detection signal are downscaled as shown in Figure 9a, wherein all blue areas indicate the spatial distribution of irrigated WW and blue shades indicate the intensity of irrigation. Downscaling normalization results eliminates the effects of non-irrigated pixels and directly expresses the spatial distribution of WW. Figure 9b,c are the results of the irrigated area provided by GIAM and GRIPC, respectively. The largest irrigated area is shown in Figure 9c because the data are classified into only four categories for agricultural areas, and the irrigation area cannot be effectively distinguished, whereas the irrigation area of the two crop rotations is shown in Figure 9b, which is close to the irrigation area identified in this paper. In recent years, due to the problem of overexploitation of groundwater in the NCP, many regions no longer plant high-water-consumption crops, such as WW, which results in Figure 9a irrigated areas being less than that of the GIAM data. Compared with the traditional irrigated area identification results, the proposed method can also reflect the irrigation intensity of the study area.

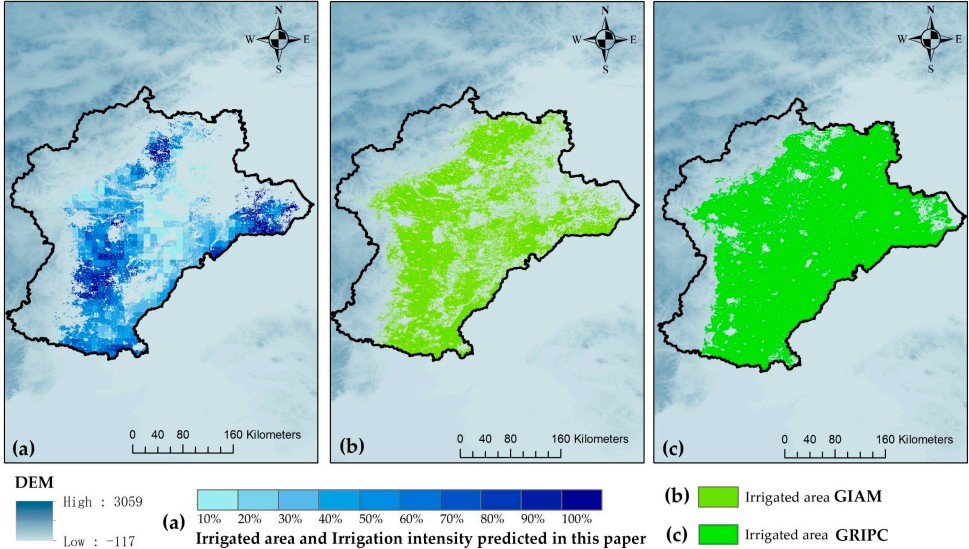

**Figure 9.** Irrigated area distribution in the study area. (**a**) shows the downscaled irrigated area and irrigation intensity results, (**b**) shows the irrigated area from GIAM, and (**c**) shows the irrigated area from GRIPC.

*4.3. Validation and Growth Consistency Analysis*

The detection results of irrigation signals in this paper will be validated from two aspects: 1) Validate the time of irrigation according to irrigation record; 2) count the consistent samples of WW growth and validate the effectiveness of the irrigation signal detection result.

**Table 2.** Overall accuracy of the irrigation timing detection results.

| | WW 1 | | WW 2 | | WW 3 | | WW 4 | | RF 1 | | RF 2 | | RF 3 | |
|---|---|---|---|---|---|---|---|---|---|---|---|---|---|---|
| | Rec | Det | Rec | Det | Rec | Det | Rec | Det | Rec | Det | Rec | Det | Rec | Det |
| Dates | | 2/26 | | 2/23 | | 2/26 | 2/26 | 2/26 | / | 3/3 | / | / | / | / |
| | 3/13 | 3/13 | 2/24 | 2/25 | 3/13 | 3/14 | | 2/27 | / | / | / | / | / | / |
| | | 3/14 | | 2/27 | 4/15 | 4/16 | | 3/3 | / | / | / | / | / | / |
| | 3/26 | 3/27 | | 3/12 | 5/10 | 5/10 | 3/14 | 3/14 | / | / | / | / | / | / |
| | | 3/31 | 3/12 | 3/13 | | | 4/10 | 4/10 | / | / | / | / | / | / |
| | 4/10 | | | 3/14 | | | 5/10 | 5/10 | / | / | / | / | / | / |
| | | | 5/11 | 5/12 | | | | | / | / | / | / | / | / |
| Accuracy | 50.00% | | 100.00% | | 75.00% | | 83.33% | | | | | | | |
| Overall accuracy | | | | | | | 77.08% | | | | | | | |

Det: irrigation detection result. Rec: irrigation records. Units marked in green indicate that the detected irrigation date matches the recorded irrigation date, and units marked in orange indicate the detection irrigation date does not match the recorded irrigation date.

In Table 2, the timing of the irrigation signal detection is compared to the timing of the irrigation records. The irrigation detection accuracy of the WW samples WW 1, WW 2, WW 3, and WW 4 used for validation were 50.00%, 100.00%, 75.00%, and 83.33%, respectively. It should be noted this irrigation record corresponds to two detection dates, with the irrigation record recorded for two days to calculate the single sample accuracy validation. Irrigation signals were also detected in the rainfed crop samples, which were added as errors to the calculation of the overall irrigation signal detection accuracy. The overall accuracy of the irrigation timing detection in this paper was 77.08%. The calculation of overall accuracy must consider the detection error of the rainfed crop region.

Since WW presents significant NDVI changes in the returning green and jointing stages and less precipitation during this period, little effect on WW growth is observed. Therefore, the returning green and jointing stages of WW are selected as the key period of growth consistency analysis. WW showed more significant growth consistency in the early stage of returning green and jointing than in other growing stages. The irrigation records show that the irrigation water used in the returning green and jointing stages is surface water, and the irrigation water in other growth stages is irrigated groundwater. Surface water irrigation is a unified supply for water resource management departments, and groundwater irrigation is privately accessible to farmers. Different irrigation times are the main reason for the inconsistency in WW growth. WW is irrigated by surface water during these two growth stages, and surface water irrigation can cover a wide range of WW regions. Irrigation caused an increase in the SMAP pixel value (SM), which was used to identify an irrigation pixel. Due to the low spatial resolution of SMAP data, the consistency of WW growth under the coverage of one SMAP pixel in this study area must be discussed. If most of the WW covered by one SMAP pixel shows a consistent increase in the NDVI, then the spatial distribution of WW is effective for downscaling the irrigation signal. Conversely, if the increase in the NDVI for most WW (covered by one SMAP pixel) is inconsistent, then the irrigation signal identified by the SMAP pixel cannot effectively express WW growth. In Figure 10, SM, NDVI changes (after upper envelop) and irrigation time for different SMAP samples were plotted. Figure 10 a–d correspond to Validate WW1, Validate WW2, Validate WW3 and Validate WW4 in Figure 7a, respectively.

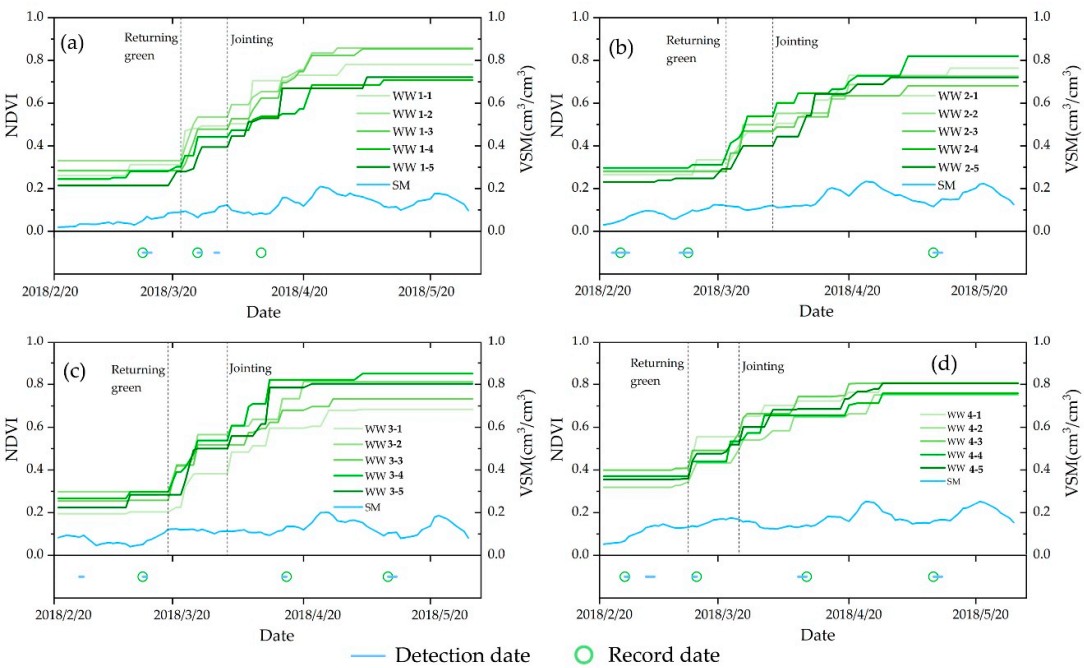

**Figure 10.** SM, NDVI changes (after upper envelop) and irrigation time for different SMAP samples.

According to the NDVI variation treatment method shown in Figure 10, 55 NDVI samples covered by 11 SMAP WW samples were validated for WW growth consistency. The number of samples with same increase trend of WW NDVI in the returning green and jointing stages was counted separately. For example, at the time of the returning green stage, the simultaneous increase in the NDVI indicates consistency among the WW growth samples, and vice versa. By counting the number of consistent WW samples covered by different SMAP pixels, the percentage of WW growth consistency covered by SMAP pixels can be calculated, and the results are shown in Table 3. In Table 3, the ratio of the consistent growth of WW covered by SMAP pixels is greater 70%, and in some regions, it can reach 100%. The overall consistency result reached 83%, and the results show that the irrigated area after downscaling can effectively express the true WW irrigation situation.

**Table 3.** Statistical results of the winter wheat sample consistencies.

|    | WW1 | WW2 | WW3 | WW4 | WW5 | WW6 | WW7 | WW8 | WW9 | WW10 | WW11 |
|----|-----|-----|-----|-----|-----|-----|-----|-----|-----|------|------|
| RG | 3 | 5 | 4 | 2 | 3 | 5 | 5 | 4 | 3 | 5 | 4 |
| J | 4 | 4 | 4 | 5 | 5 | 5 | 5 | 4 | 4 | 3 | 5 |
| P | 70.00% | 90.00% | 80.00% | 70.00% | 80.00% | 100.00% | 100.00% | 80.00% | 70.00% | 80.00% | 90.00% |
| OA | | | | | | 82.72% | | | | | |

RG: returning green stage; J: jointing stage; P: percentage; OA: overall accuracy.

## 5. Discussion

### 5.1. Comparison with Other Studies

Lawston et al. [33] proposed a method for detecting irrigation signals based on SMAP data. In this method, dates are first selected during the crop growing season and then the characteristics of SM are compared at irrigated and non-irrigated points. Finally, the method uses time integrated and SM normalized metrics of SM and precipitation to detect irrigation signals. According to the method, the precipitation and SM processing results are obtained, as shown in Figure 11a,b. Since this method does not deduct the effect of precipitation from the SM changes, in the southern part of the study area, sufficient precipitation affects the detection of irrigation signals. Simultaneously, the detection results of the proposed method are normalized, which is more conducive to the comparison of the

two methods. In Figure 11, the amplitude change in (c) is more obvious than (d), and some obvious regions in the calculation results are marked. Region 1 contains two large reservoirs adjacent to the Taihang Mountains. Region 2 is the southern part of Beijing. Region 4 is a large wetland named Baiyangdian. The type of underlying surface may affect the monitoring of time series SM changes. Notably, region 3 is the main irrigation area in the southern part of the Hebei Province. However, the irrigation signal for this irrigated area is not significant in (d). Therefore, the method proposed in this paper is more suitable for irrigation signal detection in the study area.

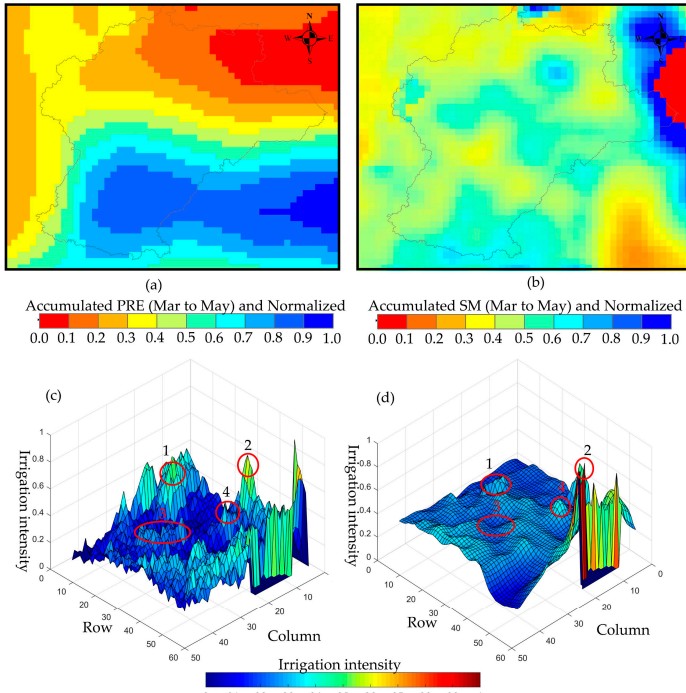

**Figure 11.** Comparison of the method proposed in this paper with the time-integrated and SM normalized irrigation signal detection methods. (**a**) Accumulated PRE and normalized result, (**b**) accumulated SM and normalized result, (**c**) irrigation intensity calculated by this paper proposed method, and (**d**) time-integrated and SM normalized irrigation signal detection methods. Both normalized results and irrigation intensity are dimensionless variables.

The method proposed in this paper can acquire daily irrigation signal detection result, so the research can describe the irrigated information in the study area in more detail. In order to display the irrigation details more abundantly, the monthly irrigation signals were accumulated to acquire a monthly distribution of WW irrigation (as shown in Figure 12). At the end of February, the southern part of the study area warmed up, and the irrigated area of WW was mainly concentrated in the southern part. In March, a wide range of WW was irrigated, and irrigation in April and May was concentrated in the central and western regions. Compared with existing studies, Chen et al. statistically analyzed the climate distribution characteristics of WW growing season in the NCP for many years, which is consistent with the monthly spatial distribution of irrigation in this paper [50]. Yang et al. collected information on crop planting and irrigated area in the NCP for many years, and acquired crop and irrigation spatial distribution characteristics in this region [51]; the results of Yang's study are similar to the results acquired in this paper, but due to the change of crop pattern in the eastern region, inconsistencies have been caused. Overall, the results of this paper are consistent with existing research findings.

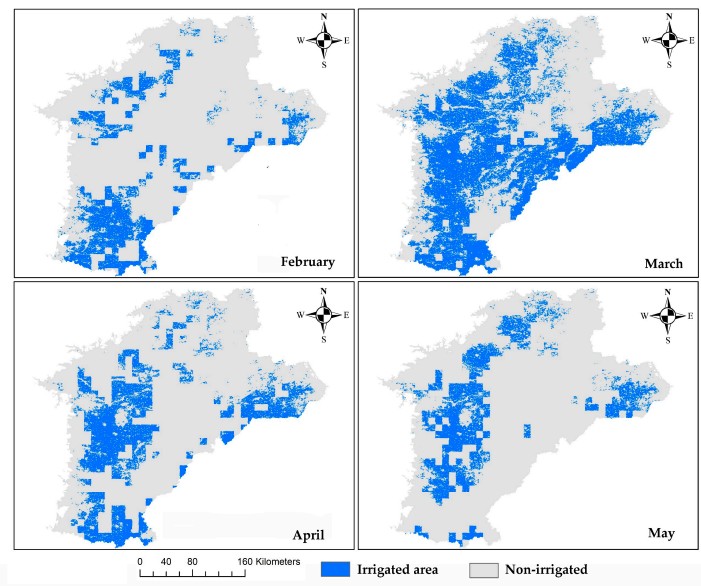

**Figure 12.** Spatial distribution of winter wheat irrigated area.

In the study of irrigated area extraction without considering SM changes, most of the research extraction methods are based on time series vegetation index changes and supervised classification to identify irrigated areas [13,34,41]. These methods for identifying irrigated areas through optical remote sensing datasets were based on identifying the type of crop to distinguish whether the area is irrigated [52]. The common advantage of these methods is that they can obtain a high resolution crop spatial distribution, and the accuracy can be increased as the spatial resolution of remote sensing images increases, and validated in many areas [16]. Based on the SMAP data extraction irrigation signal, the spatial distribution of crops with high spatial resolution is was introduced as irrigation reference area, which not only maintain irrigation time and frequency information, but also increases precision of SMAP data recognition irrigated area.

### 5.2. A Rational Discussion of the Irrigation Signal Detection Model

The establishment of the irrigation signal detection model in this paper was based on irrigation records and SMAP SM data. Since the SM change data in the irrigation record is measured every 10 days, the data does not express time-continuous SM variations, so the SMAP SM data is not compared with the in-situ data. However, validation of SMAP SM data and irrigation-induced SM increase researches can demonstrate that variations in SMAP SM data can be used to establish irrigation signal detection models. A validation study of SMAP SM data has been described in the first section, and this section will discuss the relationship between irrigation and SM variation.

Chen et al. [23] analyzed the continuous variation of SM before using the MODIS Greenness Index to detect irrigation signals in Gansu Province. Combined with irrigation and precipitation records, it was found that the sudden increase of SM generally originated from irrigation and effective precipitation. At the same time, the irrigation time was estimated using the continuous SM variation data in year 2016. Under the condition of no in-situ SM data, Lawston et al. [33] obtained the SMAP SM variation of different crop types based on the location information of irrigation and rainfed crops, and according this, they extracted the irrigated area of many regions in the United States. Is the phenomenon of SM sudden increase caused by irrigation also obvious in the NCP region? Some studies based on the effects of different irrigation patterns on WW yield provide a reliable basis. Wang et al. [53] collected SM variation in different irrigation patterns of winter wheat. The data show that although the SM (soil depth 0–80 cm) covered by WW in drip irrigation is slightly lower than level-basin, there is obvious SM increase after WW irrigation. Zia et al. [54] collected more detailed time series SM variation data (soil depth 10 cm and 40 cm); at the soil depth of 10 cm,

irrigation will cause significant SM increase, while at 40cm, irrigation will maintain a higher level of SM, and the sudden increase is not significant. In this study, when the in-situ SM data is insufficient, the SMAP SM data can be used to analyze the SM variation characteristics of WW and rainfed crops. Referring to number of studies on the relationship between irrigation and SM response, this paper suggests that irrigation records and SM increase can be used to detect irrigation signals in agricultural areas.

It should be noted that the thresholds in the irrigation signal detection model proposed in this paper are not universal. For example, in the study area of this paper, there are significant differences in SM increase caused by different irrigation patterns. In areas with more complicated irrigation patterns, the irrigation pattern of sample points needs to be considered. In addition, the SMAP SM data of 9 km resolution is acquired by 36 km data downscaling, and the uncertainty of scale conversion may also affect the application of the model. If necessary, consider using multiple filtering methods for data optimization.

## 6. Conclusions

Based on multisource remote sensing data, including SMAP, MODIS, and an irrigation map, the 5-point moving average method was used to detect irrigation signals in southern Hebei. Then, irrigation record data were used to validate the accuracy of the irrigation signal detection results. The accuracies of the four WW samples used for validation are 50.00%, 100.00%, 75.00%, and 83.33%, and the overall accuracy is 77.08%. The consistency analysis of 55 WW growth samples showed that the growth consistency of WW reached 82.72% in two large-scale surface water irrigation areas. Based on a consistency analysis, the downscaling method can be used to downscale the WW irrigation signal detected by the SMAP data. The proposed irrigation signal detection and downscaling method are more suitable for the detection of large-scale surface water irrigation signals. Limited by the spatial resolution of SMAP data and continuous in situ measured SM data, small-scale groundwater irrigation signal detection is difficult to establish. In future research, small-scale groundwater irrigation signal detection will be further studied.

**Author Contributions**: Z.H. and H.Z. designed the experiment; C.Z. and Y.J. provided revisions to the paper; H.W. and H.Z. provided financial support; and Z.H. wrote the manuscript.

**Funding:** National Key R&D Program of China (Grant No. 2018YFC0407705), Fundamental Research Funds for the China Institute of Water Resources and Hydropower Research (WR0145B012017), and Fundamental Research Funds for the China Institute of Water Resources and Hydropower Research (WR0145B272016).

**Conflicts of Interest:** The authors declare no conflicts of interest.

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
