# Peer review of "Detecting Winter Wheat Irrigation Signals Using SMAP Gridded Soil Moisture Data"

_remotesensing, doi:10.3390/rs11202390_

Round 1
Reviewer 1 Report
The manuscript has some interesting content and has attempted to address some of the interesting features soil moisture (SMAP) to be applied for irrigation application. The manuscript is very sound in terms of the dataset used and the research questions set up. However due to the poor organization and deviation from the main research question, the manuscript is hard to follow. The main question of the manuscript is how the limitation (poor spatial resolution) of SMAP SM products (with good temporal resolution) can be improved by integrating some other phenological and statistical irrigation data or the insitu soil moisture data is very interesting and the research results should be centered on this topic.
Some of the major issues of the manuscript
Although the main objective of the paper is soil moisture retrieval from SMAP to detect irrigation, I found a significant portion of the manuscript was loaded with identification of winter wheat with the rainfed crops. This makes the central crux of the shadowed and disrupt the flow of the manuscript. Both in the methods and results section identification of WW area has been a priority. In fact, a single graph with the NDVI/EVI will give the phenology signal and ET threshold can give the indication of irrigation. Since authors argued these two parameters thresholds are obvious for identifying irrigated winter wheat then the question of using SMAP to further advancement should be the topic of the manuscript. Redundancy is the issue throughout the manuscript.In many places in the introduction where authors try to prove a point on the spatial and temporal resolution. Please summarize effectively like" Despite having the low spatial resolution, SMAP provides high temporal resolution SM products. To address the spatial resolution issues, this study ...............". Similarly, in the research results the redundancy is in explaining the identification of WW areas with the non-winter wheat areas. I would suggest, don`t start with one topic and bring another topic in middle and again go back to the previous topic. Each paragraph should have only one topical sentence. Only the information that brings in depth knowledge of the main research questions needs to be discussed to a great extent and other supportive topics should be summarized concisely so that the readers are not distracted from the main story. I have a major issue in the methods section. The issue is not the technical one but the clarity. There are so many thresholds used to identify different parameters it is very hard to follow. I strongly suggest to come up with certain kind of presentation like a flow chart example decision tree format- if this threshold is met go to next or something like that, otherwise its hard to get the grasp of the methods used. Another major issue is the discussion. There is no any references in the discussion and the introduction of the new ideas in the discussion (towards the end) add confusion. The discussion should be developed based on the findings of the result section, giving more illustrations of the findings discussing with the references from literature so that the findings from the current study is more credible and the readers will know what gap existed in the study subject has been added from the current study. Some other comments: The title is too long. In my opinion, the title should reflect the central message of the manuscript. I would prefer title like " detecting wheat irrigation signal using SMAP gridded soil moisture data " L17-19: hard to understand. Rewrite the sentence. L24:29: the abstract do not highlight the major findings of the research rather focused on explaining the details of the methods L35:"with the development of society"- delete the phrase L41- is crop irrigation is the main cause of over exploitation of ground water? how about the use in industries/ livestocks or household uses? "overexplotation" are two terms L42:safety or sustainable production? the second term is more appropriate L43: This is very confusing. Do the authors want to detect the signal of irrigation ( after irrigation is applied) or want to detect signals from vegetation health status as a signal for applying irrigation? The area and frequency terms here giving the intuition of the detection of irrigated wheat crop and how many times the crop has been irrigated, while the timing give the impression that based on crop health it is the time to irrigate wheat.The paragraph should be framed to give the clear picture of what is the main question of the study. Upto this point I am still not clear what problem is the manuscript addressing? L98:99-What was the issue that they failed to detect? be more descriptive L153:"everywhere on earth's surface" - replace with "at global scales" L155:abrupt end of the statement L161-162: this necessary? what value does this add to the manuscript?. The whole message can be summarized simply something like this;The study used the SMAP SM datasets from the end of Feb to the end of May (2015-2018) corresponding to the winter wheat irrigation period in study area" 167-168:I did not see the representation of these 8 control points in the map in Fig.1 L236:Obvious is too general term. delete the obvious characteristic and use the statement in the parenthesis L240:Any references to claim this? L254-255:This is repetition,. earlier March NDVI was stated as the characteristic feature L315:Use past tense because the work has been already done L350: Delete "results" L365: Please change the order of the graphs putting WW sites together L371-373: Revise, long sentence with unusual endingL377-378: Do any sources have the statistical data for irrigated winter wheat area? The irrigation schedule (which month/day) should be presented so that it can give a hint on increased ET due to irrigation. I assume WW is not irrigated throughout the growing season. In fig 6, I saw the shaded area as the irrigated period. Is irrigation continue for a 20 days or even a month, are those all sites have the same irrigation period? In non-irrigated WW also, we observed increased in ET as the growing season progress towards the peak. L391: "precipitation of different crops"- rewrite this L394-396:Here I found the irrigation schedule but still are those same for all the samples you have chosen? How the variability in start and end of irrigation between the sample sites have been addressed? L401: So for building the models this variability needs to be addressed. So I would suggest, to use the sample site specific irrigation dates to test /build the models. One or two case study can be discussed. Overall, the discussion section needs more improvement but the manuscript is not ready for further comments/review at this present form.Author Response
Dear reviewer
Thank you for your very valuable comments, we have revised every comment you provided, and we think your comments are very professional and detailed. We have provided a revised manuscript for you, and you can review each of our corrections.
One thing that needs to be specifically explained is that we have re-plotted Figure 6. The irrigation period in the original image is the time range over which irrigation has existed for many years. To reduce ambiguity, we plotted the irrigation time for each training sample in Figure 6. In fact, we used the sample site-specific irrigation dates to test /build the models.
Based on your comments, we also adjusted the structure of the article and significantly revised the discussion section.
Thank you for reviewing again.

Reviewer 2 Report
Your manuscript (obviously revised after preceding review) is presenting the utilization of most progressive remote-sensing data for a pressuring problem not only in Hebei Province, in a very qualified and perfect manner. I found but very minor details to be improved, such as missing coordinates in Fig. 1 or rethinking to give the percent numbers with two decimal places (meaning a really outstanding significance). Also a short definition of the EASE-grid could be given for the less-initiated reader. However, all in all, your manuscript constitutes a publishable contribution for Remote Sensing.
Author Response
Dear reviewer
Thank you for your very valuable comments, we have revised every comment you have provided, and we think your comments are very professional and detailed.
Thank you for reviewing again.

Reviewer 3 Report
SUMMARY OF THE REVIEW
The submitted paper aims to detect winter wheat irrigation signals based on SMAP enhanced soil moisture downscaled with MODIS remote sensing data. Herein, Authors have made a great contribution to the SM research studies, several technical aspects of the study were implemented perfectly and explained sufficiently. Undoubtedly, authors invested huge amount of time and have made a great effort to produce this high-quality of research which is clearly structured; the language used largely appropriate. I would like to congratulate warmly the authors for producing this high-level research paper, and as final recommendation, I see that this manuscript in its form and level DESERVES TO BE ACCEPTED FOR PUBLICATION after considering the MINOR COMMENTS that I pointed below.
DETAILED COMMENTS.
The title is adequate for the content of the paper but I suggest the following
“Detecting winter wheat irrigation signals based on SMAP enhanced soil moisture downscaled with MODIS remote sensing data”
The abstract is perfectly written. The introduction provides good background to the research topic and delivers the objectives of the work correctly. In the second part (study area) and the third part of this paper (mat and methods), put the figures after the text.
Please make sure to define ALL THE ACRONYMS from their first appearance in your paper. All references MUST BE CHECKED and formatted as required by MDPI, also make sure that all the references have DOI number unless it is not available.
Author Response
Dear reviewer
Thank you for your very valuable comments, we have revised every comment you have provided, and we think your comments are very professional and detailed. We have provided a revised manuscript for you, and you can review each of our corrections.
Thank you for reviewing again.
